# Defining the Sharing Economy for Sustainability

**Steven Kane Curtis \***  **and Matthias Lehner**

The International Institute for Industrial Environmental Economics (IIIEE), Lund University, Tegnérsplatsen 4, 223-50 Lund, Sweden; matthias.lehner@iiiee.lu.se

**\*** Correspondence: steven.curtis@iiiee.lu.se; Tel.: +46-046-222-0216

**Abstract:** (1) Background: The sharing economy has emerged as a phenomenon widely described by academic literature to promote more sustainable consumption practices such as access over ownership. However, there exists great semantic confusion within academic literature surrounding the term "sharing economy," which threatens the realisation of its purported sustainability potential. (2) Objective: The aim of this paper is to synthesise the existing academic definitions and propose a definition of the sharing economy from the perspective of sustainability science in order to indicate sharing practices that are consistent with the sustainability claims attributed to the sharing economy. (3) Methods: We conduct a database search to collect relevant academic articles. Then, we leverage qualitative content analysis in order to analyse the authors' definitions and to synthesise the broad dimensions of the sharing economy in the discourse. (4) Results: We propose the following characteristics, or semantic properties, of the sharing economy for sustainability: ICT-mediated, non-pecuniary motivation for ownership, temporary access, rivalrous and tangible goods. (5) Conclusion: The semantic properties that inform our definition of the sharing economy for sustainability indicate those sharing practices that promote sustainable consumption compared to purely market-based exchanges. This definition is relevant for academics studying the sustainability impacts of the sharing economy in order to promote comparability and compatibility in research. Furthermore, the definition is useful for policy-makers, entrepreneurs, managers and consumers that have the sharing economy on the agenda in order to promote social enterprise and support sustainable consumption.

**Keywords:** sharing economy; sustainability; literature review; interdisciplinarity

---

## 1. Introduction

In the Special Report of the IPCC released in October 2018, scientists warn that our window for preventing catastrophic climate change is closing [1,2]. This report is just the latest of a growing chorus of warnings. According to scientists, the over-exploitation of natural resources as a result of our unsustainable consumption, especially in more-developed countries, is the cause of this catastrophic collapse of animal species [3] and contributes to continued greenhouse gas emissions [2]. Our unsustainable consumption results in habitat loss, biodiversity loss, pollution and climate change, among other impacts [2,3].

As such, solutions to address our unsustainable consumption patterns are needed. Strategies to dematerialise our economies permeate academic discourses including product-service systems [4], access-based consumption [5], collaborative consumption [6,7], the sharing economy [8], among others. However, the practical applications and subsequent sustainability implications of sustainable consumption approaches seem incremental in the face of our growing sustainability challenges [9].

Therefore, in order to meaningfully address these sustainability challenges, more needs to be done in order to realise the purported sustainability potential of alternative modes of consumption,

such as the sharing economy. The sharing economy is largely promoted in academic literature as offering access over ownership [10,11], by leveraging the idling capacity of goods and services [12,13], in order to reduce our overall consumption and subsequent resource use [14,15].

Widely, the sharing economy is promoted by practitioners, industry associations, policy-makers and academics because its purported sustainability potential. However, the sustainability impact of the sharing economy remains understudied [16], especially considering rebound effects [17–19].

Furthermore, the academic discourse surrounding the sharing economy offers inconsistent conceptualisations of the phenomenon depending on the discipline and focus of research. Some suggest that this leads to semantic confusion surrounding the sharing economy, which permeates academic and popular discourses [20,21]. For actors interested in the sharing economy because of its purported sustainability potential, the lack of a consistent definition as to which types of practices and organisations belong under the umbrella of the sharing economy may lead to undesirable outcomes (e.g., for managers and practitioners) [22]. Furthermore, the phenomenon may be co-opted or exploited in ways that the purported sustainability potential is not realised [23], with some authors claiming the "[sharing economy] is simply a faster and up-to-date form of market economy" [24] (p. 4269). Already, the term 'share-washing' is used to describe exploitative economic ventures that operate under the "warm glow" of the sharing economy umbrella [25,26].

We suggest that this semantic confusion has a negative impact on current and future perception of the sharing economy, threatening the potential for the sharing economy to mainstream. As such, researchers, policy-makers and entrepreneurs interested in institutionalising sharing as a consumption practice may wish to be concerned with the increasing negative sentiment among consumers, described in popular science publications. For example, in a 2017 US national survey conducted by the National League of Cities, 51% of those surveyed had mixed feelings about the sharing economy [27]. Moreover, the sharing economy is described as leading to reduced equity and justice as a result of casualised labour markets and financialising of housing [28]. Finally, using Uber as an example, a recent *Guardian* article called for "tougher rules" governing the sharing economy [29].

We fear the semantic confusion within academia—which also permeates among policy-makers, entrepreneurs, managers and consumers—hinders the institutionalisation of sharing as a consumption practice and threatens the realisation of the purported sustainability potential of the sharing economy at scale needed to address our grand sustainability challenges. As such, our aim is to synthesise the existing academic definitions and propose a definition of the sharing economy from the perspective of sustainability science in order to indicate the types of practices that may lead to more sustainable outcomes. Our definition is intended to support academics, policy-makers, entrepreneurs, managers and consumers that promote the sharing economy for its purported sustainability potential.

We approach our analysis from the critical realist ontology and the discipline of sustainability science, an interdisciplinary research field, which seems appropriate in order to integrate conceptualisations among social and natural science disciplines. Of course, we appreciate that other disciplines study the sharing economy from different perspectives, including from management and economics. However, these conceptualisations must be logically consistent and at least have the potential to deliver on its purported sustainability potential if authors continue to promote the sharing economy in this way.

In the following sections, we will provide an overview of the sharing economy as a phenomenon. Then, we will describe our methods, which resulted in the dimensions of the sharing economy that capture the breadth of the discourse in academic literature. Then, we analyse these dimensions, seeking to align logically the discourse to arrive at semantic properties to propose a definition of the sharing economy for sustainability. Finally, we discuss the implications of our proposed definition in how the sharing economy has been conceptualised thus far.

## 2. Sharing and the Sharing Economy

The meaning of sharing has been previously discussed by past academics including Price [30], Belk [31] and John [32]. Depending on the context, sharing could mean to share: as an act of division into equal parts; as an act of distribution; as a form of common ownership; as an act of communication; or as a form of individual expression online [32]. Throughout the English language, the word 'sharing' has shifted in meaning and continues to do so, especially with its prolific use on social media.

Considering 'sharing' and 'economy,' the words do not obviously relate to each other. Nonetheless, the term 'sharing economy' has emerged. The sharing economy is described as "a rising pattern in consumption behaviour," experiencing "immense growth" [33] that "is surpassing any other markets in outlook and market growth" [12]. The practice of sharing promises many societal benefits: to provide an opportunity to save and/or make money [34,35]; to change consumer behaviour [12,36]; to reduce resource use and usher in more sustainable consumption [13,37]; to facilitate sustainable economic growth [38]; and to enhance social cohesion in cities [39–41].

Quite broadly, the sharing economy as described by literature includes a variety of consumption practices and organisational models. Among authors and across disciplines, there is great disparity in the types of activities that are described within the sharing economy (Table 1). In sampling a diversity of definitions, we see tensions emerge: the extent of online mediation (e.g., online vs. offline); whether the exchanges allow transfer of ownership (e.g., renting vs. donating); the role and place of money (e.g., pecuniary vs. non-pecuniary motivation); and, the actors involved in the exchange (e.g., peer-to-peer vs. business-to-consumer).

**Table 1.** Diverse definitions of the sharing economy from literature.

| Source | Definition |
| --- | --- |
| Aloni, E. (2016) | " . . . an economic activity in which web platforms facilitate peer-to-peer exchanges of diverse types of goods and services" [25] (p. 1398) |
| Barnes, S. & Mattsson, J. (2016) | " . . . involves access-based consumption of products or services that can be online or offline" [42] (p. 200) |
| Cheng, M. (2016) | " . . . describes the phenomenon as peer to peer sharing of access to under-utilised goods and services, which prioritizes utilization and accessibility over ownership, either for free or for a fee" [43] (p. 111) |
| Habibi, M.R., Davidson, A., & Laroche, M. (2017) | " . . . non-ownership forms of consumption activities such as swapping, bartering, trading, renting, sharing and exchanging" [22] (p. 113) |
| Hamari (2016) | " . . . the peer-to-peer-based activity of obtaining, giving or sharing the access to goods and services, coordinated through community-based online services" [44] (p. 2047) |
| Heinrichs, H. (2013) | " . . . individuals exchanging, redistributing, renting, sharing and donating information, goods and talent . . . " [13] (p. 229) |
| Shaheen, S., Chan, N.D., Gaynor, T. (2016) | " . . . a popularized term for consumption focused on access to goods and services through borrowing and renting rather than owning them" [45] (p. 165) |

As a result, scholars describe the sharing economy as an umbrella term [13,22,46], covering a variety of behaviours and business models that cannot be narrowed down to one specific definition [47–49]. Moreover, scholars often treat the term 'sharing economy' as synonymous to related terms such as 'collaborative consumption' [50]. These competing conceptualisations leads to semantic confusion in the research field and beyond [20,22,47].

The sharing economy is described as a neologism [32,51], which may account for this purported semantic confusion. Neologisms are defined as "form-meaning pairings . . . that have been manifested in use . . . but have not yet occurred frequently and are not widespread enough in a given period to have become part and parcel of the lexicon of the speech community and the majority of its members" [52] (p. 31–32). Neologisms are described to undergo the process of institutionalisation through three stages: creation, consolidation and establishment [52,53]. When a new word or phrase is created, the

meaning is often highly ambiguous and reliant on context for correct interpretation [52]. The transition from creation to consolidation is difficult to demonstrate empirically but is often characterised by decreased semantic ambiguity as the term increases in usage [52]. The term is described as established or institutionalised with the addition of semantic properties generally known and understood by a speech community, independent of context [52].

Ambiguity persists as the characteristics, that is, semantic properties, overlap with other upcoming modes of consumption such as collaborative consumption, access-based consumption, gig economy, platform economy, among others. This suggests that the sharing economy is likely in the process of consolidation as its use permeates and efforts are made to reduce the surrounding ambiguity.

Herbert & Collin-Lachaud [47] suggest three reasons for the difficulty in arriving at a definition of the sharing economy: (1) the practices described within the sharing economy "are extremely varied, flourishing, constantly changing and subject to the fad effect"; (2) consumers and/or organisations that engage in these varied practices do not see themselves as part of the sharing economy; and, (3) out of pragmatism, stakeholders choose not to impose criteria on what constitutes the sharing economy. To be clear, scholars have embarked on conceptualising the sharing economy [10,48,54–56]; however, these conceptualisations are not developed to support the realisation of the purported sustainability potential attributed by academics and businesses to the sharing economy.

The sustainability challenges that we face as a society are urgent and mounting. Practices that promote sustainable consumption and production are required. The semantic confusion surrounding the sharing economy allows for purely market-based exchange, without improved sustainability outcomes. Therefore, this paper conducts qualitative content analysis to synthesis the existing literature in order to develop characteristics, or semantic properties, that prioritise the sustainability potential often attributed to the sharing economy.

## 3. Materials and Methods

In order to synthesise the academic discourse pertaining to the definition of the sharing economy, we conducted a literature review followed by qualitative content analysis using a grounded theory approach. Methods presented by Keathley-Herring et al. [57], Randhawa et al. [58], Corbin & Strauss [59] and Bazeley & Jackson [60] primarily supported us in developing our approaches to both.

### 3.1. Method for Data Collection—Database Search

The literature review collected academic, peer-reviewed journal articles, in particular, as this type of literature is described to facilitate broad understanding and to illuminate nuances among authors within a research field [61]. In addition, the coverage of Web of Science and Scopus databases favours academic journals over books, conference proceedings and reports within the social sciences and humanities [62].

Similar to Keathley-Herring et al. [57], we conducted a scoping study to support the identification of keywords to be used in our subsequent database search of Scopus and Web of Science. We chose the search query "'sharing economy' OR 'collaborative consumption'" to account for the fact that some authors used the terms interchangeably. Then, we included in our scoping study the ten most cited articles in Scopus and Web of Science, which yielded 18 articles after accounting for duplicates. These articles were thoroughly read by two researchers to distil other related terms to the sharing economy. The scoping study arrived at thirty-eight related terms (Appendix A), which were used to execute the subsequent database search relating to the sharing economy. The database search was conducted on 10 May 2017 and included articles starting 1 January 1978; results were limited to academic, peer-reviewed journal articles written in English. The search results returned 2270 articles, including duplicates between databases, 869 from Scopus, 1401 from Web of Science (Appendix A). The titles, keywords and abstracts were reviewed to confirm relevance for this study based on exclusion criteria determined by our scoping study (consult Appendix B). The review yielded 272 articles from Scopus and 213 articles from Web of Science (485 including duplicates). Of these, we initiated a second

round of review by removing: articles that were duplicates between the databases; articles in which we could not obtain full access as PDF; or, articles incorrectly catalogued based on type of literature, language or contradicting metadata between the databases. Furthermore, we chose to remove articles that mentioned the 'share economy' exclusively, as this concept is intellectually unrelated to the sharing economy discourse. The 'share economy,' originally conceptualised as a profit-sharing scheme to tackle stagflation, was first written about by Weitzman [63] and later rebutted by many scholars throughout the 1980s and early 1990s. Presumably, its inclusion in the literature stream stems from its similar root word and several authors conflating the term with sharing economy [13,64]. Our final sample included 255 academic, peer reviewed journal articles that make up the basis for next stage of analysis.

### 3.2. Method for Data Analysis—Qualitative Content Analysis

Qualitative content analysis (QCA) is a method to systematically study the meaning of qualitative data (i.e., language). A core feature of QCA is the development of a coding framework, either theory-driven or data-driven [65]. In line with a grounded theory approach, we use data-driven strategies to develop our coding framework inductively [59,65]. Our data-driven coding framework was developed through processes of open, axial and selective coding.

Open coding, sometimes referred to as in vivo coding, uses words or short phases directly from the text to assign as codes [65–67]. As the codes become increasingly dense, the codes are grouped into relevant categories and sub-categories [66]. Axial coding begins to understand links between these categories, yet still flexible to new codes and categories as one continues textual analysis [65,66]. Finally, selective coding arrives at core categories that are further refined and integrated into the emerging theory [65].

We used NVivo 11 for Mac, developed by QSR International, to aid in analysis of our sample articles. Computer-assisted qualitative data analysis software (CAQDAS), such as NVivo, is particularly useful when working with large amounts of textual data to engage in the above coding processes in a systematic way [65]. Furthermore, NVivo is a useful tool to conduct literature reviews [60,68–71], especially when engaging in an analytical task [60], such as synthesising the definitions of the sharing economy across academic literature. Furthermore, NVivo seeks to reduce human error during the coding process as well as analyse the data across multiple categories [71].

We proceeded with two phases: (1) we identified the definitions provided of the sharing economy in all articles within the final sample; (2) we coded all definitions of the sharing economy to arrive at the dimensions of the sharing economy, which describe the broad categories discussed in literature. Continuously, throughout both phases, we utilised features in NVivo, such as memos and annotations, to begin to interrogate and analyse the data.

3.2.1. Phase One

The unit of analysis, that is the object of investigation, was the definition of the sharing economy in each academic article. As such, our first phase was concerned with coding for the relevant definition in each academic article. In NVivo, one codes the unit of analysis as cases [60]. For example, Cheng [43] (p. 111) states that the "[s]haring economy describes . . . peer to peer sharing of access to under-utilised goods and services, which prioritizes utilization and accessibility over ownership, either for free or for a fee." This text would be coded as a case labelled 'sharing economy.'

However, some authors conflated terms, in part, due to the continued semantic confusion. For example, Barnes & Mattsson [42] (p. 200) state: "Collaborative consumption is embedded within the "sharing economy," which involves access-based consumption of products or services that can be online or offline." In these instances, the text would be coded to multiple cases including 'sharing economy,' 'collaborative consumption' and 'access-based consumption.'

We used a predatory reading approach, focusing more attention on the relevant parts of the text that may contain the authors' stated or theorised definition such as abstract, introduction, literature review (where relevant) or conclusion. We developed and discussed iteratively a Phase One Coding

Protocol based on our purpose for content analysis (see Appendix C). To begin, we each coded the same fifteen articles to ensure inter-coder reliability. Using NVivo, we calculated our strength of agreement measured by the kappa statistic. The calculated kappa statistic was 0.66, which suggests substantial agreement among coders [72]. After discussions and refinement of our coding protocol, we proceeded coding the articles individually. After coding for cases in all 255 articles, relevant definitions were coded to the case 'sharing economy' in 151 of the articles (Figure 1).

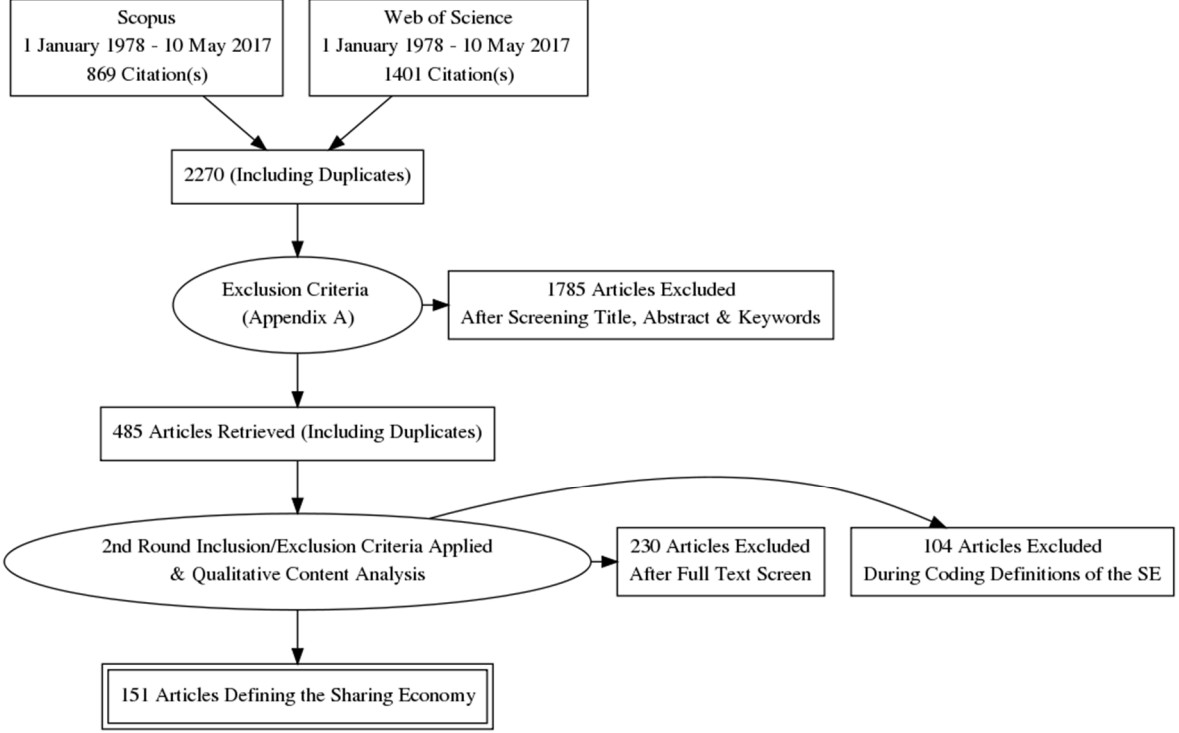

**Figure 1.** Flow Diagram: Screening of Articles Defining the Sharing Economy.

### 3.2.2. Phase Two

Once we completed coding for cases, we proceeded with analysing the definitions of the sharing economy. Once again, we developed and discussed iteratively a Phase Two Coding Protocol (see Appendix D). We applied a grounded theory approach using the processes of open, axial and selective coding.

*Open Coding*

To begin, we conducted line-by-line coding, using words or short phrases in each line to form new codes. We open coded the definitions in 34 of the 151 articles; we began to reach saturation of some codes. At that point, we had developed 821 codes. In several sessions, we grouped all codes that appeared connected and established higher order categories and related subcategories. We used the 'Mind Map' function in NVivo to visualise our emerging coding framework and generate parent and child nodes in NVivo for continued coding.

*Axial Coding*

Using the emerging coding framework, we continued coding line by line, assigning codes to either an existing category or creating a new code. In doing so, we proceeded to forge links between and within categories, as new codes continued to better describe the emerging categories. Once again, after coding 64 of the 151 articles' definitions, we became satisfied with the density of many of these core categories and subcategories. At this point, we confirmed our coding framework and proceeded with selective coding.

*Selective Coding*

We proceeded coding the remaining 87 of 151 articles' definitions using the coding framework established through the processes of open and axial coding. This included refining categories; we revisited categories by condensing or expanding their focus. We sought to define each core category and sub-category through relevant empirical data to provide weight and context to the emerged codes.

Once all of the definitions were coded, we congregated once more to discuss and explain the core categories. The core categories are said to be "broad and abstract enough to integrate the other categories and to cement the components of the phenomenon" [66] (p. 1276). We describe these core categories as dimensions of the sharing economy, which we report in Section 4. From these dimensions, in Section 5, we propose characteristics that support our definition of the sharing economy for sustainability.

## 4. Findings

The identified dimensions of the sharing economy represent the wide breadth of characteristics coded within the 151 articles in which definitions of the sharing economy were coded. They are not normative; the dimensions do not constitute a definition of the sharing economy. Instead, they are descriptive of the discourse and demonstrate the semantic confusion within literature. Relevant to the development of characteristics that inform a definition, we report on the following dimensions: motivation, ICT-enabled, idling capacity, platform or organisational models, shared goods and services as well as ownership.

For each dimension, we report the categories that emerged from our data (e.g., ICT-enabled → online) and, where possible, provide the number of articles in which we found the category to occur. Furthermore, we also include the total number of times—i.e., the frequency—the category occurred, representing the intensity of discussion.

### 4.1. Motivation

Literature suggests the proliferation of the sharing economy is driven by the great recession of 2007–2008 [16,73–77], growing social inequality [73,78], increased environmental awareness [14,16,74,79,80], proliferation of ICT [44,74,81] and convenience [45,79,82–85], among others. As such, authors expressed differing motivations among actors involved in the sharing economy. Based on our data, we found these motivations to include economic, environmental and social motivation.

Economic motivation is amorphously discussed across the literature. Our analysis seeks to structure the purported economic motivations; we coded economic motivation from the perspective of users, providers, businesses and the community. Users within the sharing economy are said to be motivated by access to a greater number of goods and services [49,86,87], which are less expensive than ownership [34,45,79,80,88–90]. By increasing users' access to goods and services at a reduced price, users have the ability to access things they otherwise could not afford, leading to increased purchasing power elsewhere in the economy [14,77]. Likewise, providers see the sharing economy as a means to generate extra income [35,82,91,92].

Businesses see the economic potential in the sharing economy; in an often-cited report by PwC, the sharing economy is said to have generated USD $15 billion in revenue globally in 2015, increasing to USD $335 billion in revenue by 2025 [12,82,93–96]. More recently, in China, according to official figures, it is reported that the sharing economy generated $500 billion USD in transactions among more than 600 million users in 2017 [97]. By 2020, officials predict that the sharing economy may account for more than 10% of China's gross domestic product (GDP), increasing to upwards of 20% by 2025 [97]. While the true scale of the economic potential is difficult to measure (especially without a widely-accepted definition to understand and compare the types of consumption practices included in the revenue estimates), literature describes the sharing economy as reducing barriers to entry [86] and introducing new business models for generating profit [98]. It is purported that this business potential will lead to increased employment [43,93,99] and more resilient communities [93,100]. Based on this, some authors propose that the sharing economy is guided by economic maximisation over

altruism [5,45,101,102]. However, a study conducted by Bucher, Fieseler, & Lutz [89] showed that economic motivations ranked third behind social-hedonic and moral motivations.

Beyond the prevailing economic motivation, consumers are said to be socially driven to participate in the sharing economy [103]. They are said to be seeking more meaningful social experiences beyond the traditional business-consumer paradigm [81,104,105]. Harmaala [12] suggests the sharing economy is an antidote to the isolating nature of social media and digitisation. Furthermore, the sharing economy is said to reduce social inequality by allowing for a more equitable distribution of goods and services [50,89].

However, widely, the literature in our sample promotes the sharing economy on the basis of its purported sustainability potential. Literature frames users in the sharing economy being driven by increased environmental awareness and the purported sustainability potential of the sharing economy [14,16,44,74,79,81,106]. The purported sustainability potential lies in leveraging the idling capacity of goods and services to reduce net consumption, which leads to reduced resource use [14,15]. It is also claimed that sharing leads to a reduction of water and energy use [38,93] and a reduction in consumer waste [15,73,84]. Ultimately, it is claimed that the sharing economy leads to reduced greenhouse gas emissions [15,17,93,106], positioning sharing as an alternative consumption practice to address climate change [12,44,76,107].

While environment and sustainability are discussed broadly, some authors suggest that sustainability aspects are merely co-benefits associated with sharing, although, there are more important utilitarian motivations among users underlying the consumption practice [18,108]. For example, convenience is a significant motivating factor not to be overlooked [18,45,79,80,83–85], achieved through the mediation of providers and users leveraging ICT.

## 4.2. ICT-enabled

Although not universal, the sharing economy is largely described as being 'ICT-enabled.' While authors indicate that sharing is not a new phenomenon [6,91,99,106], the 'newness' of the sharing economy seems to stem from the use of technology to facilitates the efficient mediation or exchange between users and providers [10,44,48,73,84,100], creating a two- or multi-sided market [109–113]. In turn, this reduces the transaction costs associated with sharing among strangers [109,114]. Technology helps to reduce these transaction costs by:

- Improving access to information [113,115–117]
- Facilitating intermediation between providers and users [46,48,87]
- Facilitating payments [90,93]
- Facilitating a reputation or review system [116–118]
- Increasing convenience [79,119]

The underlying explanatory factor for this increased economic efficiency in our dataset is primarily linked to technological innovation. While offline sharing exists, many sharing organisations utilise some form of technology. The internet, smartphones, social media and algorithms are discussed with respect to platform intermediation, bringing about the efficiency gains ascribed to the sharing economy.

Our analysis demonstrated that numerous terms are used in literature to describe the extent to which the sharing economy is ICT-enabled (Table 2). In total, these terms are used in 108 of the 151 article definitions coded in our sample literature.

The diversity of terms used to describe the means and extent of ICT-enabled intermediation of the sharing economy adds to the semantic confusion. For example, authors use the terms 'internet' and 'web' interchangeably; however, there is an important distinction: the internet describes networking infrastructure whereas the web describes the information-sharing model that is built on top of the internet. Furthermore, contrasting digital and smart technologies, both 'digital' and 'smart' focus on ICT integration whereas 'smart' also acknowledges the connection to systems, infrastructure and prescribes an underlying environmental motivation [120]. The breadth of discussion in literature

pertaining to these dimensions leads to questions regarding the type, form or extent of ICT-enabled mediation taking place in the sharing economy and the subsequent impact on realising purported sustainability outcomes.

**Table 2.** ICT-Enabled—Number of Articles and Frequency of Appearance in Dataset.

| Term | Articles | Frequency |
|---|---|---|
| Online | 58 | 138 |
| Internet | 48 | 74 |
| Technology | 45 | 82 |
| Smartphone | 16 | 20 |
| Smart Phone App | 14 | 24 |
| Social Media | 12 | 26 |
| Website | 11 | 13 |
| Social Networks | 7 | 9 |
| ICT | 7 | 9 |
| Internet-based | 6 | 6 |
| Mobile Technology | 6 | 6 |
| Web 2.0 | 6 | 7 |
| Online Community | 5 | 5 |
| Online Platform | 5 | 5 |
| Virtual | 5 | 9 |
| Internet-mediated | 4 | 4 |
| Technology-driven | 4 | 4 |
| 3rd Industrial Revolution | 3 | 3 |
| Big Data | 3 | 3 |
| Digital Platform | 3 | 3 |
| Mobile Device | 3 | 3 |
| Online Marketplace | 2 | 2 |
| Web Platform | 2 | 2 |

*4.3. Idling Capacity*

The sharing economy is said to leverage the excess, surplus or underutilised nature of idling goods and services [12,13]. In defining the sharing economy, authors distinguish it from other forms of consumption based on leveraging the idling capacity of goods or services to facilitate *access*. Goods and services in the sharing economy are said to have high idling capacity [108], which describes the percentage of time the goods or services are being utilised. Our analysis identified several terms that describe this idling capacity: excess, excess capacity, idle, intense use, latent, spare, spare capacity, surplus, unproductive, underused and unused (Table 3).

**Table 3.** Idling Capacity—Number of Articles and Frequency of Appearance in Dataset.

| Term | Articles | References |
|---|---|---|
| Idle | 18 | 25 |
| Spare | 13 | 19 |
| Unused | 12 | 15 |
| Underused | 10 | 11 |
| Excess | 9 | 11 |
| Surplus | 9 | 11 |
| Underutilised/Under-utilised | 9 | 15 |
| Excess Capacity | 8 | 8 |
| Latent | 2 | 8 |
| Space Capacity | 2 | 3 |
| Unproductive | 2 | 3 |
| Intense Use | 1 | 1 |

Idling capacity, using some formulation of the above words, appears in 60 of the 151 definitions contained within our dataset. As many of the definitions are limited or incomplete, we find this is a dominant dimension of the sharing economy. However, the exact formulation and scope of idling capacity is varied across our sample. For example, considering the category 'idle,' which appeared most frequently in our sample, authors discussed 'idle' in relation to resources [12,34,98,111,121], assets [122], cars [123,124], accommodation [125] and time [92]. Similar diversity in formulation permeates the dimension.

Furthermore, we see this dimension most closely associated with the sustainability potential of the sharing economy described by many authors. Leveraging the idling capacity of goods, in particular, increases the intensity of use leading to a reduction in the need to produce new goods, which reduces the overall environmental impact associated with production and consumption.

### 4.4. Platform or Organisational Models

Within literature, the language used to describe the entity that facilitates sharing is contested. Described as a platform or an organisation, the choice by authors has implications in how the sharing economy is understood, implemented and regulated. While our analysis did not find any discussion on this, we suggest differences exist in how organisations and platforms operate pertaining to their goals, approaches and practices.

We choose to adopt the terminology 'platform' to describe the entity that facilitates sharing, in part, due to the ICT-mediated nature of the sharing economy in connecting providers and users. As such, we suggest a platform describes the entity responsible for mediating interactions between providers and users; the platform model describes the constellation of these actors involved in the interaction forming the two- or multi-sided market. Central to the platform models, *access* to goods and services, in contrast to ownership (see Section 4.6), seems a central feature. Table 4 illustrates the platform models generated as a result of coding our sample.

**Table 4.** Platform Models—Number of Articles and Frequency of Appearance in Dataset.

| Term | Articles | Frequency |
| --- | --- | --- |
| Peer-to-Peer | 64 | 211 |
| Business-to-Consumer | 10 | 15 |
| Crowd | 6 | 7 |
| Consumer-to-Consumer | 5 | 11 |
| Business-to-Business | 4 | 9 |
| Business-to-Peer | 1 | 1 |
| Public-to-Citizen | 0 | 0 |

The platform models most prominently discussed in literature include business-to-consumer and peer-to-peer models. Discussed less frequently, business-to-business models are also found within the literature. Furthermore, our coding generated three other platform models: business-to-peer, crowd and public-to-citizen. In a business-to-peer model, the platform mediates a two-sided market connecting businesses with idling assets to users in order to gain access to goods and services. The 'crowd' model describes mediation from one to many, from many to one or from many to many. In our analysis, we conceptualised 'crowd' platforms to include crowdsourcing, crowdfunding, cooperatives and shared-ownership models. Finally, public-to-citizen model describes government-maintained or supported sharing platforms.

Based on the coding of the literature within our study, the 'business' in the business-to-consumer model is the provider/merchant, providing and facilitating access among consumers to a stock of goods [98]. In this model, the business retains ownership of the goods offered in the interaction. The models are often described in relation to product-service systems [126–128] and carsharing [16]. In these examples, the business-to-consumer model does not lead to a two- or multi-sided market. Furthermore, literature describes the business-to-consumer model as the standard or traditional

commercial dyad [122], which is being challenged by other models emerging that promote more prosocial exchanges.

In contrast, the peer-to-peer platform model, appearing most frequently in our study, leverages ICT to mediate a two- or multi-sided market between peer providers and peer users. Literature also discussed consumer-to-consumer models [84,98,129], which was not fundamentally different from the peer-to-peer model. Similar in nature, business-to-business platforms also create two- or multi-sided markets, although between businesses in need of idling resources niche to their business sector. A feature of these models is that sharing takes place among equals.

The business-to-peer model occurs so infrequently in our sample, we suggest this model is not part of the dominant discourse. Furthermore, in our sample, business-to-peer is used to describe similar platforms as the business-to-consumer model; for example, literature used the example of Zipcar [25,91], which is a commercial carsharing company, to exemplify the business-to-peer model. Instead, we propose the business-to-peer model describes a platform mediating exchanges between businesses with idling resources and peers desiring access to such resources. There were no examples or discussions in our sample literature as of yet matching this model; however, we see this as a potential platform model that is within the spirit of the sharing economy.

We conceptualise and define the platform model 'crowd' drawing on literature describing crowdsourcing, crowdfunding, filesharing, cooperatives and shared-ownership models. Crowd platforms see multi-sided markets created and mediated using ICT. In contrast to peer-to-peer or business-to-business platform models, which mediate exchange typically between one provider and one user, crowd platform models mediate interactions between multiple actors. Crowd platform models connect many to one (e.g., crowdfunding), one to many (e.g., filesharing) and many to many (e.g., shared-ownership models).

Finally, some literature discussed public transportation, parks and roads as part of the sharing economy [8,12] as well as municipal-supported or operated bike sharing schemes and tool libraries [11,121]. We conceptualised this model as public-to-citizen as the provider is often a municipal government providing access to goods and services to all citizens within their jurisdiction.

*4.5. Shared Goods & Services*

The literature largely distinguished shared objects on the basis of tangible and intangible objects [11,17,46,98,105,109,130]. Based on our analysis, we categorised tangible objects to include space, durable goods and non-durable goods; in contrast, intangible objects include services, time, knowledge, money, thoughts and online content (i.e., filesharing, streaming services, photos).

In considering intangible objects, we observed services discussed in literature relating to the gig economy [10,41,131] and product-service systems [13,128]. Furthermore, intangible objects discussed in literature include time banks [48,132], crowdfunding [38,41,131] and knowledge-sharing [41,131,133]. Furthermore, authors have described streaming services such as Netflix and Spotify to belong to the sharing economy [82,84]. It is unclear if these manifestations constitute sharing, especially in contrast to the sustainability motivations prevalent in literature.

In contrast, tangible objects are described broadly as goods [18,46,101,109], assets [11,17,80] or resources [77,130]. In our analysis, we categorised these as physical goods, which encompass space, durable goods and non-durable goods. Firstly, sharing of space is described widely in literature as a room [11,87,88], apartment [73,88], home [106,114,134], accommodation [24,135], office [74,103] or parking spot [136]. Furthermore, durable goods include those goods that do not quickly deteriorate through use. We suggest that these goods also often possess high idling capacity. Examples in literature include cars, bikes, luggage, sporting goods, consumer electronics, home improvement products, furniture and homeware, among others. Finally, non-durable goods include those goods that typically have a lifetime less than two to three years, for example, clothes, personal care products and food.

*4.6. Ownership*

Authors claim there is less need and desire for ownership, especially among millennials [5,24,77,80,137]. However, there is a lack of consistency in how researchers discuss ownership of goods and services, depending on the type of goods or services being included in the authors' conceptualisation of the sharing economy.

Many authors see the sharing economy as promoting exchanges that do not lead to the transfer of ownership [18,94,95,138]; in this spirit, the sharing economy is said to facilitate "access over ownership" [10–12,101]. In contrast, other authors describe the sharing economy as facilitating redistribution and second-hand exchanges [13,18,121,139], which would constitute transfer of ownership.

As such, our analysis looked at the verbs used by authors to describe the exchanges taking place within the sharing economy to explore the notion of transfer of ownership. When describing transfer of ownership, authors include consumption practices such as bartering, buying second hand, donating, exchanging, gifting, redistributing, swapping or trading. In contrast, when describing no transfer of ownership, authors include consumption practices such as accessing, borrowing, collaborating, hiring, lending, renting, sharing, using or utilising (Table 5).

**Table 5.** Ownership—Number of Articles and Frequency of Appearance in Dataset.

| Term | Articles | Frequency | Transfer of Ownership |
|---|---|---|---|
| Access | 79 | 206 | No |
| Exchange | 61 | 190 | Either |
| Rent | 39 | 74 | No |
| Collaborate | 37 | 72 | No |
| Trade | 24 | 47 | Yes |
| Lend | 23 | 27 | No |
| Swap | 19 | 24 | Either |
| Barter | 15 | 15 | Yes |
| Borrow | 14 | 29 | No |
| Gift | 14 | 22 | Yes |
| Redistribute | 11 | 18 | Yes |
| Lease | 8 | 10 | No |
| Shared Use | 7 | 7 | No |
| 2nd-Hand | 4 | 8 | Yes |
| Share | - | - | No |
| Use | - | - | No |

In our analysis, access emerges as the dominant mechanism in the sharing economy; however, 'share' and 'use' were excluded from this discussion as they occur in too many contexts not relevant to our analysis. Furthermore, practices that promote no transfer of ownership also dominate in our analysis.

## 5. Defining the Sharing Economy for Sustainability

The above dimensions represent the breadth of the academic discourse. The initial tensions elaborated in Section 2—the extent of online mediation; transfer of ownership; the role and place of money; and, the actors involved in the exchange—are embodied again within these dimensions. However, further logical inconsistencies emerged. Firstly, the sharing economy is widely described as potentially leading to more sustainable consumption, by leveraging idling capacity and/or facilitating access over ownership. However, the conceptualisations, as well as the examples used to illustrate the sharing economy, fail to achieve one or both of these promises. Redistribution and second-hand markets clearly exhibit transfer of ownership, although these practices leverage idling capacity. Commercial carsharing platforms provide access to vehicles in contrast to ownership; however, they do not leverage idling capacity of existing vehicles.

Furthermore, some authors position the actor facilitating sharing as an organisation, others as a platform. This distinction has implications on the role of ICT and the extent it is leveraged to create two- or multi-sided markets. It is these inconsistencies, we sought to harmonise the disparities within each dimension in order to reduce the semantic confusion and promote a definition of the sharing economy, motivated by literature, that may offer more sustainable outcomes. We purpose the following semantic properties, the elements of a term that contribute meaning, that inform our definition of the sharing economy for sustainability: ICT-mediated, non-pecuniary motivation for ownership, temporary access, rivalrous and tangible goods.

*ICT-Mediated*: The sharing economy is mediated by ICT, creating two- or multi-sided markets. The fundamental 'newness' of the sharing economy, as compared to traditional forms of sharing, is the mediated exchange enabled by ICT. This leads to reduced transaction costs through the provision of information, access to new markets and facilitating payment. While we view that the exchange may take place either online or offline, it must also be mediated by technology, either formally (e.g., app or website) or informally (e.g., Facebook group).

*Non-Pecuniary Motivation for Ownership:* The sharing economy leverages the idling capacity of goods. If the sharing economy is to leverage idling capacity, as widely discussed in literature, it follows that the goods shall not be purchased or owned only for the purpose of making money through sharing. We do not see it as inconsistent for the sharing economy to involve monetary or other forms of compensation. However, while providers (and platforms) may make money from sharing, the owner presumably owns the good for their own use, leveraging its idling capacity when not in use through the sharing economy.

*Temporary Access:* The sharing economy is characterised by consumption practices that do not lead to transfer of ownership. Although not unanimous, throughout our sample, the sharing economy is said to facilitate "access over ownership." It is distinguished from other forms of consumption such as buying second-hand, swapping, donating, trading or gifting because sharing does not lead to the transfer of ownership (although sometimes it does may lead to shared-ownership). Furthermore, as the sharing economy is said to leverage idling capacity, it follows that access, not transfer of ownership, increases the intensity of use of the goods shared. We clarify "access over ownership" further to describe temporary access, which is differentiated from renting over longer periods of use such as rental cars or apartments.

*Rivalrous:* When sharing, the use of shared goods prevents the simultaneous use by another. Some literature discusses public transit, parks and roads as examples of the sharing economy. However, we see goods that are accessible simultaneously by all as poor examples of the sharing economy, in part, because they possess low idling capacity and unlimited access. Therefore, we suggest shared objects shall be rivalrous. This criterion subsequently eliminates filesharing and video streaming, among others, as examples of the sharing economy because they are non-rivalrous exchanges.

*Tangible Goods:* The sharing economy sees sharing of space, durable goods and nondurable goods. Following the logic of the above semantic properties, the sharing economy shall facilitate exchange of tangible goods, in contrast to intangible goods. First, tangible goods possess clear idling capacity. Second, tangible goods possess clear mechanisms for access and ownership. Finally, tangible goods are rivalrous.

## 6. Discussion

The above semantic properties inform our definition of the sharing economy for sustainability. However, they have some implications on how the sharing economy has been previously conceptualised. Figure 2 illustrates the semantic properties and the subsequent activities excluded from our definition, which are sometimes attributed to the sharing economy.

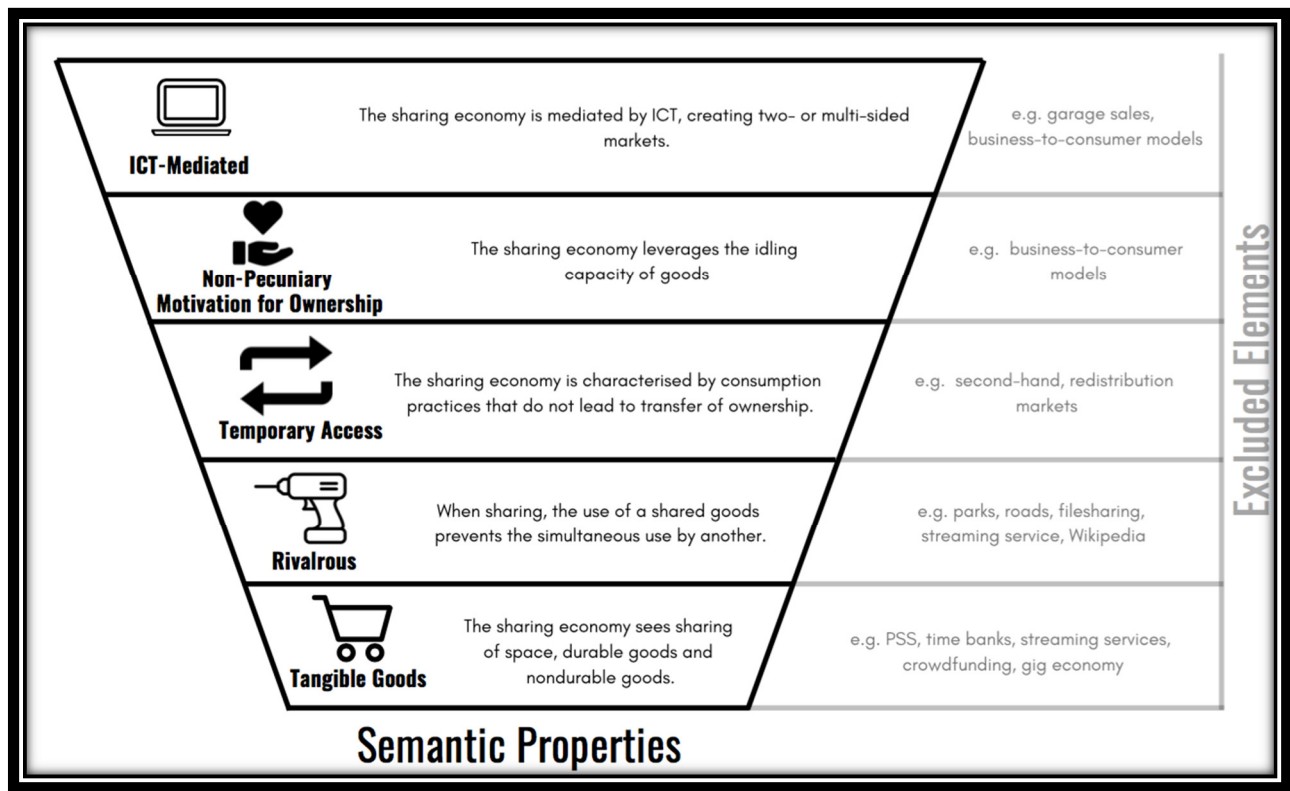

**Figure 2.** Semantic Properties of the Sharing Economy and Excluded Elements.

Below, we discuss the following implications on the conceptualisation of the sharing economy: (1) excludes business-to-consumer models; (2) excludes second-hand and other redistribution markets as well as other exchanges that allow for the transfer of ownership; (3) excludes intangible objects such as services, time, money, knowledge, streaming content, among others; and (4) we propose consumables (e.g., food, personal care products) may be included in the sharing economy, despite transfer of ownership.

Firstly, we suggest that business-to-consumer models be excluded from the sharing economy. Our semantic properties position the sharing economy as ICT-mediated. The distinction between the sharing economy and other forms of consumption that also leverage ICT is the creation of two- or multi-sided markets connecting providers and users to facilitate access of idling tangible goods. Therefore, in our definition, we argue business-to-consumer models are excluded from the sharing economy as ownership of the goods is retained with the business that facilitates the exchange. This fails to constitute the creation of a two- or multi-sided market. As such, many carsharing (e.g., Zipcar, DriveNow, Car2Go) and bike sharing (e.g., Donkey Republic, Ofo) schemes fall outside of our proposed definition, provided the assets are not peer-owned.

Furthermore, we also suggest that business-to-consumer models are excluded as pecuniary motivation drives ownership among the suppliers in the model; therefore, suppliers do not leverage the idling capacity of goods. Instead of sharing, we suggest this type of exchange is more closely aligned with existing literature on product-service systems [4] or access-based consumption [5]. In instances where businesses do provide access of idling goods to users, unless mediated by a platform, we suggest this exchange is more closely aligned with existing literature on collaborative consumption [6].

However, we do not go as far as some [43,56] to say that sharing only takes place between peers. While it is likely that the semantic properties proposed manifest in peer-to-peer exchanges, they may also include business-to-business, business-to-peer and crowd platform models. These models may leverage ICT to create multi-sided markets and may provide temporary access of idling tangible goods.

Secondly, our definition excludes exchanges that allow for the transfer of ownership including second-hand or other redistribution markets. While these models likely lead to more sustainable consumption, we suggest these models extend product lifetimes, slowing resource loops but lead to a transfer of ownership. As such, we suggest second-hand markets and other redistribution markets, which are included in several conceptualisations of the sharing economy, more closely align with existing literature on the circular economy.

Next, we argue that the sharing economy facilitates the sharing of tangible objects in contrast to intangible objects, which are excluded from our definition. While an individual may provide a service as a result of idling time and acquired (owned) knowledge, we argue that these types of exchanges are better described by the literature pertaining to the gig economy [140] and do not embody the purported sustainability motivations described in our sample. Other types of intangible objects may possess idle capacity such as knowledge, thoughts, files or streaming content; however, while ownership of intangible objects is often possible, these do not have a clear mechanism for maintaining or transferring ownership. Furthermore, intangible objects are often not rivalrous, in that multiple people can access these objects at the same time (e.g., Wikipedia, Netflix, Spotify, Twitter).

Finally, we propose that consumables, such as food and personal care products, are included in the sharing economy, despite requisite transfer of ownership. This is as long as the other semantic properties, namely, ICT-mediation and idling capacity, are fulfilled. This is because consumables are characterised through one-time use and, as such, transfer of ownership is inherent in the use of these products. It is simply impossible to return food once eaten or put a spritz of perfume back in the bottle. However, we distinguish consumables from non-durable goods (e.g., clothes), which are described to have a lifespan less than two to three years. As such, in keeping with the above semantic properties, access over ownership to clothes and other non-durable goods is expected. While this excludes second-hand shops or Really Really Free Markets [141], it captures clothes libraries, tuxedo

or dress rentals as well as recreational equipment and other rental services mediated online in a two- or multi-sided market.

## 7. Conclusions

The sharing economy has the potential to realise more sustainable consumption practices; however, at present, the semantic confusion surrounding the sharing economy detracts from realising this potential. We seek to better define the sharing economy, especially with regard to sustainability, in order: (1) to indicate those practices that may lead to more sustainable outcomes, in comparison, to those that are associated with purely market-based exchange, deemed "share-washing"; (2) to promote the institutionalisation of sharing as a sustainable consumption practice; and, (3) to support the comparability of research examining the economic, social and environmental impact by academics, especially when used as decision-support among policy-makers and businesses.

The aim of this article was to synthesise the existing academic definitions of sharing economy and propose a definition of the sharing economy from the perspective of sustainability science. Guided by literature, we propose a logically coherent definition consisting of relevant semantic properties: ICT-mediated, non-pecuniary motivation for ownership, temporary access, rivalrous and tangible goods. These characteristics seek to harmonise the purported potential of the sharing economy among diverse conceptualisations born from different disciplines. Our definition can help indicate those sharing practices within the sharing economy that prioritise sustainable consumption. This is particularly relevant for academics studying the sharing economy from the perspective of sustainability as well as policy-makers, entrepreneurs and consumers interested in the sharing economy for its sustainability potential.

We acknowledge that our derived definition is not the only possible way to conceptualise the sharing economy. Researchers in diverse disciplines are motivated to study the sharing economy broadly from their own interests and perspectives. We acknowledge this right. However, should authors continue to suggest that the sharing economy may lead to sustainable outcomes, their conceptualisations must at least have the potential to deliver on its purported sustainability potential. If we, as academics, fail to be critical of the sharing economy and its current implementation due to continued semantic confusion, we suggest the sustainability potential of the sharing economy may never be realised.

Limitations: Our study was conducted from the critical realist ontology and the discipline of sustainability science. As such, readers should interpret our analysis and the suggested definition with this in mind. Furthermore, as with any literature review, articles that are chosen for inclusion or exclusion influence the findings. We have chosen to only include academic peer-reviewed journal articles written in English and justified this approach. Furthermore, the articles in our study are static and do not take into account advancements in knowledge since our initial database search in May 2017. However, since then, we have monitored publications and believe our article contributes to the research community.

Future Research: Through our analysis of literature, we recorded over forty different terms used to describe similar consumption practices (e.g., sharing economy, collaborative consumption, platform economy, access-based consumption). We propose these terms used to describe a variety of consumption practices (e.g., sharing, renting, buying) belong to a semantic domain. A semantic domain describes a set of words or concepts that possess common semantic properties [142]. One approach used in semantics when words and concepts have shared semantic properties is called distinctive feature analysis [142]. Future research may wish to explore the semantic properties possessed by each term and distinguish these competing terms through distinctive feature analysis. For example, access-based consumption may be similar to the sharing economy except it does not harness idling capacity. This approach may further support the reduced semantic confusion within the semantic domain and support the institutionalisation of alternative sustainable consumption modes.

Furthermore, discourse analysis may be another useful method for those interested in studying the construction of meaning among different stakeholder groups, for example, users or the political or power relations among actors involved in the sharing economy. The choice to use discourse analysis should consider how researchers " . . . conceptualise subjectivity, the structure of meaning and the processes that produce that meaning" [143] (p. 6); however, different approaches can be considered along three dimensions; ontology; focus; and purpose [143]. In particular, this method might be relevant when examining the definitions of the sharing economy based on the position in society from which the actor comes from or the purpose and motivation for the actor to define the sharing economy in that way. Of course, this can illuminate who is setting the agenda, for what purpose and how this might impact the institutionalisation of the sharing economy.

**Author Contributions:** Conceptualisation, S.C., M.L.; methodology, S.C.; software, S.C., M.L.; formal analysis, S.C., M.L.; resources, S.C., M.L.; data curation, S.C.; writing—original draft preparation, S.C.; writing—review and editing, S.C., M.L., O.M.; visualization, S.C.; supervision, O.M.; funding acquisition, O.M.

**Funding:** This research has received funding from the European Research Council (ERC) under the European Union's Horizon 2020 research and innovation programme (Grant Agreement No. 771872) and Riksbankens Jubileumsfond (The Swedish Foundation for Humanities and Social Sciences) (Grant Agreement No. RIK16-1055:1).

**Acknowledgments:** We would like to thank Oksana Mont (O.M.), Yuliya Voytenko Palgan and Andrius Plepys for their comments and support in the writing process.

**Conflicts of Interest:** Authors declare no conflicts of interest. The funders had no role in the design of the study; in the collection, analyses or interpretation of data; in the writing of the manuscript or in the decision to publish the results.

## Appendix A. Database Search Results

**Table A1.** Search Terms and Returned Results.

| Search Term | Scopus | Web of Science |
|---|---|---|
| "business model" AND sharing | 291 | 268 |
| entrepreneurship AND sharing | 153 | 554 |
| "sharing economy" | 147 | 105 |
| "sharing economies" | 147 | 3 |
| PSS AND sharing | 64 | 108 |
| "collaborative consumption" | 55 | 47 |
| "the mesh" AND sharing | 51 | 159 |
| "peer-to-peer exchange" OR "P2P exchange" | 28 | 18 |
| "product-service system" AND sharing | 28 | 24 |
| "product service system" AND sharing | 28 | 24 |
| "share economy" | 26 | 23 |
| "resource pooling" AND sharing | 19 | 25 |
| "sustainable consumption" AND sharing | 19 | 58 |
| "social innovation" AND sharing | 18 | 29 |
| prosumer AND sharing | 18 | 15 |
| "social lending" | 18 | 9 |
| "collaborative economy" | 16 | 10 |
| "access-based consumption" | 15 | 12 |
| "sharing model" AND commercial | 6 | 1 |
| "prosocial sharing" | 5 | 4 |
| "alternative consumption" AND sharing | 4 | 4 |
| "extended self" AND sharing | 4 | 19 |
| "commercial sharing systems" | 3 | 2 |
| "collaborative lifestyle" | 3 | 1 |
| "market-mediated access" | 2 | 2 |

**Table A1.** *Cont.*

| Search Term | Scopus | Web of Science |
| --- | --- | --- |
| "fractional ownership" AND sharing | 2 | 3 |
| "alternative marketplaces" AND sharing | 1 | 1 |
| "peer-to-peer trading services" OR "P2P trading services" | 1 | 0 |
| "result-oriented services" AND sharing | 1 | 1 |
| "non-exchange-based sharing" | 0 | 0 |
| "sharing model" AND non-commercial | 0 | 0 |
| "non-ownership-based sharing" | 0 | 0 |
| "redistribution markets" AND sharing | 0 | 0 |
| "non-reciprocal ownership" AND sharing | 0 | 0 |
| "direct-contact collaboration" | 0 | 0 |
| "system-hookup collaboration" | 0 | 0 |
| "segregated collaboration" | 0 | 0 |
| "use-oriented services" AND sharing | 0 | 0 |
| Sum | 1173 | 1529 |
| Duplicates between searches | 304 | 128 |
| Total | 869 | 1401 |

## Appendix B. Motivation and Exclusion Criteria

Produced and agreed upon as of May 11, 2017 during joint discussions among two researchers on the basis of reading the 18 articles that made up the scoping study.

The motivation of the study is to examine the ways in which sharing has been described as new as a part of the sharing economy. This is done by using qualitative content analysis to construct the dimensions of the sharing economy or at least differentiate the sharing economy from other similar models (access-based consumption, redistribution markets, commodity exchanges, gift-giving, product service systems, bartering, lending, renting, leasing, etc.). As such, the following exclusion criteria guide our selection of articles that form our final sample for analysis. Only the article title, keywords and abstract are reviewed.

We exclude articles on the basis of the following:

1.  Results that are not academic articles
2.  Articles not indexed as written in English
3.  Articles that do not include any of the search terms in the Title, Abstract or Keywords
4.  Articles that do not appear to be relevant, at least in part, to discussing the concept of sharing
5.  Articles that describe sharing as an act of communication (ex. knowledge sharing, information sharing, information as pricing) or as an act of individual expression online (social media)
6.  Articles that discuss the sharing of data/files, unless it is rivalrous and peer-to-peer (i.e., distributed shared memory)
7.  Articles that describe risk sharing or profit and loss sharing, unless it is discussed as part of the Share Economy (concept by Weitzman)
8.  Articles that discuss peer-to-peer lending, unless it specifically mentions it as a part of the sharing economy or collaborative consumption

We chose to include electricity sharing/pooling from renewable energy, as electricity is rivalrous and has inherently environmental benefits.

## Appendix C. Phase One Coding Protocol

*21 July 2017*

Phase One: Coding for Cases

1.  Select the next unassigned article from shared Google Sheets: Final Document List

2.　From the drop-down menu, select yourself as the person 'Coded by'
3.　Within the selected article, code for cases, the unit of analysis, which includes the definition for the sharing economy and/or any other related model to the sharing economy (i.e., collaborative consumption, access-based consumption). These terms should come from the text, seeking to see how the authors use other terms to describe other but related, terms.

　　a.　Coding for existing and new cases:

　　　　i.　New cases include terms/words that the authors use that act as synonyms or variations (i.e., related models) that replace or contrast the sharing economy in the text.
　　　　ii.　When a new term relating to the sharing economy is introduced in relation to another, they should both be coded as cases (ex. The article describes the sharing economy as part of collaborative consumption. In this instance, the same text should be coded for both sharing economy and collaborative consumption).

　　b.　The content that makes up the case should include and be limited to any text within the article that describes the sharing economy (or related models) and any of its features/factors/dimensions/drivers/motivations/outputs/impacts or any content that provides additional understanding or meaning of the sharing economy or related models.

　　　　i.　It is appropriate to exclude content that the coder determines unnecessary in creating a better picture of the sharing economy or related models, in particular, detailed results and discussion from each article.

　　c.　One should also code for all examples that are expressed in the article in relation to the sharing economy (or related models).

4.　Annotations can/should be written in NVivo to document preliminary analysis of the article and its content. In particular, this may be a thought to consider moving forward based on the text or a future research question with the dataset.
5.　Memos are not necessary but can be linked to a specific article to provide greater context/description of thinking in relation to the linked article.

**Appendix D. Phase Two Coding Protocol**

*26 September 2017*

Phase Two: Coding of Cases

　　The goal of this phase is to code the case 'sharing economy' to begin to develop concepts for each of the terms used to describe the sharing economy and related activities. The process seeks to employ a grounded theory approach—namely open, axial and selective coding practices—to arrive at concepts for each of the cases. The coders will begin with processes of open coding. Once saturation is agreed upon by both coders, the coders will begin to work with the codes to arrive at a preliminary coding framework. At this point, the coders move onto processes of axial coding, seeking to code based on the preliminary coding framework but being open to codes that continue to emerge from the data.
　　The below process details the coding protocols:

1.　Code for one case at a time in alphabetical and subsequent order of articles within each case based on the Google Sheet.
2.　Select the next unassigned article from Google Sheets: Coding of Cases
3.　From the drop-down menu, select yourself as the person 'Coded by'
4.　Within the selected case and article, employ open coding processes:

　　a.　Code as nodes

b. Code entire sentences

c. Code for context, when appropriate

d. Code for all terms used to describe the particular case

5. It is at the coders' discretion for which to code for and not to code for, based on discussions among the coders and in line with the above proposed research objectives, as agreed upon by the coders.

6. Due to working collaboratively in two different NVivo files, the coders shall utilise ANNOTATIONS (instead of MEMOS as they are not include when merging files) to:

a. Preserve thinking with regard to the rational for choosing a particular code (if necessary)

b. Preserve and suggest future thinking for analysis

c. Put forward emerging concepts

7. One should also code for all examples that are expressed in the article in relation to the sharing economy (or related models).

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
