# Peer review of "Defining the Sharing Economy for Sustainability"

_sustainability, doi:10.3390/su11030567_

Reviewer 1 Report

I congratulate the authors for the chosen theme. The collaborative economy is developing rapidly around the world, I agree with the authors in the justification of the research work.
However, I believe that the paper should improve a lot. I suggest that the authors reflect and improve the following points.
The abstract explains the nature of the paper, but it would be better to explain directly the objective of the paper, the methodology used and its findings in scientific terms and also implications for management. That is, the abstract must be more concrete, to ensure that the reader gets a better idea of what he is going to read.
The authors say in the abstract: 'With a reduced ambiguity, we seek to support these actors in their capacity to understand, implement and regulate the economy of exchange to support the realization of their supposed sustainability'. This phrase is not well understood and I can not see the meaning it has in the abstract. I recommend that the authors include the originality of the work, the objective and the main findings.

The authors could include in Keywords: review of the literature and the collaborative economy concept, collaborative consumption.
The work has done a wide revision of works, although I find that some are missing such as:

N. Askitas and K. F. Zimmermann. (2009, April) Google econometrics and unemployment forecasting. Dunker & Humblot eJournals. [On-line]. Available: http://ejournals.duncker-humblot.de/doi/abs/10.3790/aeq.55.2.107
Palos-Sanchez, P.R. & Correia, M.B. (2018). The collaborative economy based analysis of demand: study of Airbnb case in Spain and Portugal. Journal of Theoretical and Applied Electronic Commerce Research 13 (3).
S. J. Barnes and J. Mattsson, Understanding Current and Future Issues in Collaborative: A Four-Stage Delphi Study, Technological Forecasting and Social Change, vol. 104, pp. 200-211, 2016.
C. Anderson, The New Industrial Revolution. New York: Crown Business, 2012.

Palos-Sanchez, P. e Correia, M. B. (2018). Impacto da economia colaborativa medida através de períodos de pesquisa na internet: relato de caso Blablacar. Revista Turismo & Desenvolvimento, 1 (27/28), 1341-1354.

J. Owyang, C. Tran and C. Silva, The Collaborative Economy. United States: Altimeter, 2013

Also, the authors must use graphics to improve the understanding of the work: such as word clouds or the VOSviewer bibliometric analysis software.

Finally, the authors have data and a very interesting study, but they should improve this paper both in quality, structure. explanation of its scientific basis and presentation of the results. I wish you good luck.

Reviewer 2 Report

1). 2.1.  I suggest a little bit more detailed title: Semantics - the words "economy" and "sharing"

2). Lines 144-147 represents a succession of quotes. I appreciate that a simple succession of the quotes is not relevant. A table with some points of view regarding the subject is preferable. At the same time, source 22 is used 4 times in one page.

3). 3.1. Literature review - represents more a matter of methodology, it does not represents literature review. I understand that the paragraph explains how the literature review was achieved. I suggest to rename it.

4). 4 - Findings - after you described the 6 dimensions, I suggest a short synthesis of them, in order to link the chapter with the next one.

Reviewer 3 Report

1-      The paper aimed to develop a definition for sharing economy. Reviewing literature, the paper attempted to address the complexity of the term and to address the ambiguity of the definition for sharing economy. The authors’ main purpose of addressing the ambiguity was “to support academics, policy-makers, entrepreneurs, managers, and consumers in their ability to understand, implement and regulate the sharing economy in order to support the realisation of its purported sustainability potential.”

2-      However, the introduction section explained that the paper specifically aimed to defend the concept of sharing economy against the academic economic approach! Reading the introduction seems the aim of the paper is not consistent with the stated aim in the abstract.

3-      The paper basically lacks critical analysis of the concept in the literature. The paper later analysed the literature to provide support for one specific assumption and prove it: economic dimension and motivation is not reliable and sharing economy offers a win-win game that everyone would take benefit from it.  

4-      the section related to the etymology of the signifiers did not assist in the provision of clear analysis and interpretation.

5-      The paper should provide a better understanding of semiology and its philosophical principles; in particular, modes of reasoning and logic of relation between phrases and concepts.

6-      In general, NVIVO provides a narrow analysis for research and intellectually the researcher(s) should develop an understanding of the relationships between signifiers and/or signified. Again a deep understanding of philosophical definitions of the signifier, signified and signification and how a signification is shaped is the task of a researcher in this area. The researchers need to read more about Claude Lévi-Strauss’ works and how semantic should be applied to analyse significations for a (floating) signifier such as sharing economy.

7-      The paper did not provide a new or convincing definition for the concept of sharing economy. It repeated obvious features attributed to the concept. The work did not present new research. It is a summary of using NVivo to count the frequency of words that lacks a critical understanding of the concept.

8-      In particular, the authors failed to understand the economic dimensions of the sharing economy. Obviously, there is a lack of enough understanding of the economy and economic definition and how the concept is related to the Economic Crises 2008. Surprisingly the authors interpreted this as “economic motivation is amorphously discussed across the literature”.

9-      The aim stated “to support academics, policy-makers, entrepreneurs, managers, and consumers in their ability to understand, implement and regulate the sharing economy in order to support the realisation of its purported sustainability potential” was not achieved at the end.

Reviewer 4 Report

Thank You very much for the possibility to become familiar with an interesting article. It is very well-written and has a theoretical character. The Authors should be appreciated for the research reliability and proper selection of wide and interdisciplinary source literature. The strong points of this article are also its layout and the clarity of presented contents. However, I think that Authors should consider the greater synthesis of the presented contents. In my opinion, the article is too extensive. For instance, Subsection 2.1, in my opinion, is unnecessary. Authors address the text to educated readers, so they do not need to explain them the basics. Table 1 should be placed in an appendix, and the results in it should be sorted in descending order.

Author Response

See attached.

Round  2

Reviewer 3 Report

Thank you for your responses.

1)       Sharing economy is currently my main research project. In addition, since I teach economics, sharing economy is an alternative mode of the market economy after neoliberalism, since 2007-8 when the neoliberal economy failed to provide balances in markets (especially in financial and property markets). Principally economic crisis is an ontological cause of the emergence of the sharing economy, it is a scientific fact and a verified statement. Sharing economy was suggested as a solution to failures in the market economy mainly by economists and global policy-makers. There is several literature that supports this argument, as the authors covered in the reference list.

2)       The first sentence of the abstract, “The sharing economy has emerged as a phenomenon widely described by academic literature to promote more sustainable consumption practices such as access over ownership” is paradoxical. First, literature showed that sustainability is an unintended or unplanned consequence of sharing economy. In addition, if you have found that the definition widely has been addressed in the literature why research regarding a new definition is required.

3)       I recommend a better reading and analysis of concepts and the subject of the paper. In the last submitted paper, instead of addressing problems, many sentences are repeated through the paper, for example, Lines 72 to 76 and abstract first lines.

4)       Naming the critical realist ontology is not enough, it should be really applied. What kind of critical realist ontology has been applied? From Bhaskar ‘s point of view or Marxian view or Charles Sanders Peirce (American school) or Lipscomb? If critical realist ontology was used then it should cover the economic and social as well as political relations that suggest a specific definition for sharing economy. Critical realism without political analysis of concepts, words and relations has no meaning. Fundamentally, semantic confusion has at least a political dimension which ignoring this provide a descriptive analysis rather than critical analysis. Furthermore, even using semantic properties is narrow and lacks a deep understanding of the approach. The paper considered a simple analysis of words and their roots. Providing a deeper analysis between words and the logics of deployment of each of them needs more knowledge about semantic and its different categories also a better reading of literature. For example, the definition of economy that was presented is reductionist, meaningless and irrelevant. We are talking about the market economy in a modern world with complicated relations. Ontologically, using different names for one “thing” or “phenomenon” is rooted in the political and power relations.

5)       The section related to technological advancement as a motivation is perfect.

6)       But the section related to ownership needs close investigation and rewriting. Fundamentally, neoliberalism worked through accumulation through dispossession from ownership particularly property especially for the middle and low-income groups. Sharing economy aimed to pacify this and to prevent social discontent. However, sharing economy potentially can create a deeper and bigger gap between upper and lower income groups. This is obvious in the case of home ownership and the gap between baby boomers and millennia. Since the suggested definition is developed based on different types of exchange, logically an etymology and/or historical investigation about the exchange is required. The definition lacks a good understanding of the economic concepts that are used. 

7)       You have collected and reviewed a high number of literature in the reference list, many of them for example, (Cockayne, D. 2016) explained how the economic crisis is the main cause created sharing economy and then a new culture emerged, as the paper mentioned later in the motivation sections. And this culture motivated using sharing economy digital platforms.

8)       An important point: Sustainability is not a science but a discipline includes different skills, professions and disciplines such as environmental science, planning, policy-making, design, engineering health science, economics.

9)      Basically, your research is a literature review and the aim is finding and addressing a gap (the main concern should be clarified) in the literature and this gap is correctly argued that is less attention to sustainability outcome of sharing economy as the unintended outcome of sharing economy. I think this gap should be addressed. Sharing economy is the foundation for the circular economy that is a novel and useful approach in economics that provide sustainability objectives.

10)   The aim of the paper stated that “is to synthesise the existing academic discourse and propose a definition of the sharing economy from the perspective of sustainability science in order to reduce semantic confusion among academics, policy-makers, entrepreneurs, managers and consumers and support the realisation of the purported sustainability potential”. I am wondering if there is any proof that your definition creates less confusion for the wide range of actors that you mentioned.

11)   The main problem is using Nvivo as the method of analysis. This software is not a good tool to understand the relations between words and terms. Applying a method of discourse analysis can reveal the relations and dynamics of a text or an article.

12)   Finally, the authors explained that economics, semantic, linguistics, other terms that they used are not in the area of their expertise and the focus is only on sustainability. These type of explanation is not a good justification for narrow analysis of the subject of this paper. If a researcher has a short and narrow understanding of a method or an approach he/she should not apply them. Even if the focus is sustainability, the discipline is not a science and a better understanding of the relationships between sustainability and other terms should be provided.

Author Response

See attached.

Round  3

Reviewer 3 Report

Thank you for your responses and corrections, I totally understand that NVivo is used for data analysis but it is not a good method for a literature review. It is a method of data analysis when we have a large number of interviews and we don't know how our interviewees/or focus groups interpret or think about some phenomena or different dimensions of a phenomenon when we are at the early stage of research and we need to take an exploratory approach to get some knowledge about something. Literature review needs more in-depth analysis. 

In addition, Fairclough presents just one type of 6 different types of discourse analysis that you mentioned in your responses. The definition and understanding that you presented about discourse analysis are narrow and reductionist. I suggest you use more discourse analysis for a literature review. Please see Glynos, J., Howarth, D., Norval, A., Speed, E. (2009). Discourse Analysis: Varieties and Methods. ESRC National Centre for Research Methods Review paper: National Centre for Research Methods (NCRM/014).  This source simply explains how discourse analysis might be helpful to deconstruct concepts such as sharing economy from a sustainability point of view. 

Author Response

Thank you for your responses and corrections, I totally understand that NVivo is used for data analysis but it is not a good method for a literature review. It is a method of data analysis when we have a large number of interviews and we don't know how our interviewees/or focus groups interpret or think about some phenomena or different dimensions of a phenomenon when we are at the early stage of research and we need to take an exploratory approach to get some knowledge about something. Literature review needs more in-depth analysis. 

NVivo is a useful tool for analysing interviews, of course. However, it is also useful for managing and organising data as well as querying and analysing qualitative text, which includes published academic articles (Bazeley & Jackson, 2013, p. 3). Furthermore, Bazeley & Jackson (2013, p. 179) state that NVivo “…facilitates the development of a literature review”. The authors (2013, p. 179) continue, “[e]mploying NVivo for working with literature is most appropriate when you are undertaking an analytical task with the literature”.

We state our aim is to analyse the definitions of the sharing economy across academic literature. We reference this source in our manuscript. This reference material is cited more than 4000 times, according to Google Scholar. It is widely used and we see it as a reputable source for developing our methods for a literature review using NVivo.

There is precedent in the journal Sustainability for using NVivo in conducting a literature review of published academic articles. Boldermo & Ødegaard
(2019, p. 6) reviewed 41 academic articles and used NVivo “…in order to summarize and analyse the findings”. Camacho-Otero, Boks & Pettersen (2018, p. 6) used NVivo to review 111 peer-reviewed journal articles deductively for their literature review. Pearce, Rodriguez, Fawcett & Ford (2018, p. 4) use NVivo to conduct a literature review of 390 articles. Similar to our study, they used a grounded theory approach to code inductively using NVivo. Finally, Escarcha, Lassa & Zander (2018, p. 3) review 126 articles using NVivo “…for further synthesis, coding, and analysis”. The authors (2018, p. 3) state, “[t]he use of [Nvivo] helped to reduce human error in data coding and analysing the content using multiple factors in the liteartures for review”. These authors also reference Bazely & Jackson (2013) when justifying their methods for the literature review using NVivo.

We have added some text in order to justify our approach in our manuscript:

We used NVivo 11 for Mac, developed by QSR International, to aid in analysis of our sample articles. Computer-assisted qualitative data analysis software (CAQDAS), such as NVivo, is particularly useful when working with large amounts of textual data to engage in the above coding processes in a systematic way [65]. Furthermore, NVivo is a useful tool to conduct literature reviews [60,68–71], especially when engaging in an analytical task [60], such as synthesising the definitions of the sharing economy across academic literature. Furthermore, NVivo seeks to reduce human error during the coding process as well as analyse the data across multiple categories [71].

In addition, Fairclough presents just one type of 6 different types of discourse analysis that you mentioned in your responses. The definition and understanding that you presented about discourse analysis are narrow and reductionist. I suggest you use more discourse analysis for a literature review. Please see Glynos, J., Howarth, D., Norval, A., Speed, E. (2009). Discourse Analysis: Varieties and Methods. ESRC National Centre for Research Methods Review paper: National Centre for Research Methods (NCRM/014).  This source simply explains how discourse analysis might be helpful to deconstruct concepts such as sharing economy from a sustainability point of view.

Thank you for this literature suggestion. We have reviewed the suggested literature and agree that it presents specific approaches to discourse analysis, which would be a worthwhile method to study the construction in meaning of the sharing economy. As such, we have suggested how future researchers may apply discourse analysis to study the sharing economy:

    Furthermore, discourse analysis may be another useful method for those interested in studying the construction of meaning among different stakeholder groups, e.g. users, or the political or power relations among actors involved in the sharing economy. The choice to use discourse analysis should consider how researchers “…conceptualise subjectivity, the structure of meaning, and the processes that produce that meaning” [143](p. 6); however, different approaches can be considered along three dimensions; ontology; focus; and purpose [143]. In particular, this method might be relevant when examining the definitions of the sharing economy based on the position in society from which the actor comes from or the purpose and motivation for the actor to define the sharing economy in that way. Of course, this can illuminate who is setting the agenda, for what purpose, and how this might impact the institutionalisation of the sharing economy.

However, this suggested research would represent a different study, based on the stated aim or purpose of the research, than the one we describe in our manuscript. Therefore, we have chosen not to embark in discourse analysis, as suggested by this reviewer. Our reading of the suggested literature advocates discourse analysis is suitable when concerned “with questions of meaning and the centrality attributed to subjects in the construction and apprehension of meaning. It is this concern with meaning and subjectivity that drives the selection of different methods or techniques in the study of discourse” (Glynos et al., 2009, p. 6). In line with our articulated ontological position, reflected in the approach to our research, we are more interested in what the text says, instead of who, how or why. Therefore, it would not be suitable to embark in discourse analysis for our stated aim, as we are not interested in how the authors individually and collectively construct meaning in how the sharing economy is understood by society. Furthermore, the reference does not indicate that discourse analysis is suitable for a literature review or suggest methods to do so.

As justified above, NVivo is an appropriate tool to conduct a literature review. Furthermore, as we stated insistently in our previous responses to this reviewer as well as in our manuscript, our aim is not to study the construction of meaning by different academics. We synthesis the academic definitions presented in literature, identify tensions – in particular, in relation to sustainability – and propose a logically coherent definition of the sharing economy for sustainability. 
